# Microflowery, Microspherical, and Fan-Shaped TiO₂ Crystals via Hierarchical Self-Assembly of Nanorods with Exposed Specific Crystal Facets and Enhanced Photocatalytic Performance

**Yi-en Du [1],\*, Xianjun Niu [1], Kai Hou [1], Xinru He [1] and Caifeng Zhang [2],\***

[1] Department of Chemistry and Chemical Engineering, Jinzhong University, Jinzhong 030619, China; xjniu1984@163.com (X.N.); houk456@163.com (K.H.); hexr123456@163.com (X.H.)

[2] Department of Chemistry, Taiyuan Normal University, Jinzhong 030619, China

\* Correspondence: duye@jzxy.edu.cn (Y.-e.D.); zhangcf301@tynu.edu.cn (C.Z.); Tel.: +86-351-398-5766 (Y.-e.D.); +86-351-288-6583 (C.Z.)

**Abstract:** In this paper, khaki titanium dioxide (TiO₂) crystals via hierarchical self-assembly of nanorods with different morphologies and specific exposed crystal facets were prepared for the first time by using a TiCl₃ treatment process in the presence and absence of morphology-controlling agents. The crystal structure, morphology, microstructure, specific surface area, and separation efficiency of photogenerated electron-hole pairs of the synthesized TiO₂ crystals were characterized. The photocatalytic and recycled performances of the synthesized TiO₂ crystals in the presence of shape-controlling agents, such as ammonium sulfate (AS), ammonium carbonate (AC), and urea, and the absence of shape-controlling agents (the obtained TiO₂ crystals were expressed as AS-TiO₂, AC-TiO₂, urea-TiO₂, and No-TiO₂, respectively) were evaluated and compared with the commercial TiO₂ (CM-TiO₂) crystals. The AS-TiO₂ microspheres with exposed uncertain facets exhibited enhanced photocatalytic activity for the degradation of methylene blue solution, which can be attributed to the combined effect of the anatase phase structure, relatively larger specific surface area, and the effective separation of the photogenerated electron-holes.

**Keywords:** titanium dioxide; exposed crystal facets; photocatalytic activity; combined effect

## 1. Introduction

Since Fujishima and Honda first discovered the photocatalytic water-splitting reaction in 1972, TiO₂ has become the most suitable and promising semiconductor material in the application of water splitting, dye-sensitized solar cells, Li-ion batteries, gas sensors, etc., because of its advantages of high physical and chemical stability, non-toxic, harmlessness, environmental friendliness, and low price [1–4]. The design and synthesis of TiO₂ with different polymorphs, morphologies, and exposed facets are key to effectively improving their practical application in the field of photocatalysis, as its photocatalytic activities depend critically on the crystal phase, morphology, size, surface area, heterojunction structure, and exposed facets of TiO₂ [5–8]. Among the four polymorphs of TiO₂ (i.e., anatase, rutile, brookite, and TiO₂ (B)) that mainly exist in nature [9–11], anatase usually exhibits the highest photocatalytic activity due to the most increased electron mobility and the lowest photogenerated electron-hole recombination in anatase and the rapid interaction between many organic molecules and anatase surfaces [2]. However, compared with the anatase phase (3.20 eV), brookite phase (3.40 eV), and TiO₂ (B) (3.10 eV), the decreased band gap energy of rutile (3.02 eV) leads to part of the photoresponse extending slightly into the visible light region, thus improving the utilization of sunlight [12]. Moreover, rutile is the most thermodynamically stable structure (anatase and brookite phases can be irreversibly transformed into the rutile phase during heating) with the advantages of a high

refractive index and good light-reflecting performance [13,14]. Therefore, it is imperative to synthesize rutile and anatase with exposed high-energy surfaces for practical application.

For example, rutile $TiO_2$ nanoflakes with exposed {110} facets were synthesized by using titanium (IV) isopropoxide as the titanium source and hydrochloric acid as the morphology-controlling agent under hydrothermal conditions, which exhibited an enhanced photocatalytic activity (91%) for the degradation of cinnamic acid [15]. Nanotubes/nanowires assembled from anatase $TiO_2$ nanoflakes with exposed {111} facets were prepared using titanium oxysulfate-sulfuric acid hydrate as the titanium source and glacial acetic acid as the morphology-controlling agent under high-temperature conditions, which exhibited an improved photocatalytic activity (17.4%) for $CO_2$ reduction to $CH_4$ [16]. Anatase $TiO_2$ microspheres assembled from ultrathin nanosheets with exposed 100% {001} facets were prepared by using potassium fluorotitanate as the precursor via an on-site precipitation hydrothermal method [17]. Anatase/brookite with tunable ratios were obtained using titanium bis(ammonium lactate) dihydroxide (TALH) as a titanium source and urea as an in situ $OH^-$ source under hydrothermal conditions [18]. Brookite single-crystal nanosheets with exposed {210}, {101}, and {201} facets were prepared using $TiCl_4$ as a titanium source, urea as an in situ $OH^-$ source, and sodium lactate as the complexant and surfactant under low-basicity conditions, which exhibited a superior photocatalytic activity toward degradation of methyl orange [19]. Anatase $TiO_2$ nanoparticles exposed to different percentages of {001} facets (5~60%) were synthesized using K-titanate nanowires as a precursor and different amounts of urea as a morphology-controlling agent [20]. Rounded anatase $TiO_2$, rod-like rutile $TiO_2$ growing along the [001] direction, and plate-like brookite $TiO_2$ with exposed {111} facets were prepared by hydrothermal treatment of the mixed solution of $TiCl_3$ and $H_2O_2$ with different concentrations under certain pH conditions [21]. Moreover, anatase $TiO_2$ nanocrystals with exposed {010}, {001}, and {111} facets and rutile $TiO_2$ nanorods with exposed {110} facets on the lateral surface were also prepared using the layered titanate as the titanium source, which exhibited good photovoltaic performance (5.14%) or superior photocatalytic activity for the degradation of organic rhodamine B (76.0%) and methyl orange (96.5%) molecules [22–24].

Herein, we report on the facile synthesis of rutile/brookite composites (AC-$TiO_2$) containing tufted microflowers self-assembled by tetragonal-shaped rutile $TiO_2$ nanorods with oriented growth along the [001] direction and irregular brookite $TiO_2$ nanoparticles with exposed {001} facets, anatase $TiO_2$ microspheres (AS-$TiO_2$) self-assembled by nanorods with exposed uncertain facets, and fan-shaped particles and microspheres (urea-$TiO_2$ and No-$TiO_2$) self-assembled by tetragonal-shaped rutile $TiO_2$ nanorods with oriented growth along the [001] direction. The preparation of microflowers, microspheres, and fan-shaped $TiO_2$ particles with different crystal forms and various exposed crystal facets using $TiCl_4$, $TiOSO_4$, TALH, etc., as raw material and urea as a morphology-controlling agent has been reported in previous literature [18–21,25]; however, the previous reports rarely used AC and AS morphology-controlling agents. Furthermore, although there are some reports on the synthesis of $TiO_2$ with $TiCl_3$ and different morphology control agents, there are few reports on the simultaneous use of $TiCl_3$ as a titanium source and urea as a morphology control agent, and few khaki $TiO_2$ samples have been prepared [21,26]. In this study, khaki $TiO_2$ crystals with different crystal forms, different morphologies, and different exposed crystal planes were prepared by simply changing the type of morphology-controlling agent, which is innovative to a certain extent. The anatase microspheres (AS-$TiO_2$) constructed with anatase $TiO_2$ nanorods with exposed uncertain facets exhibited the highest photocatalytic performance to that of the AC-$TiO_2$, urea-$TiO_2$, No-$TiO_2$, and CM-$TiO_2$ crystals, which can be attributed to its anatase phase structure, relatively high specific surface area, and the effective separation of the photogenerated electron-holes.

## 2. Results and Discussion

### 2.1. Crystal Structure, Morphology, and Exposed Facets

To identify the crystal phase and estimate the phase composition and crystalline size of the as-prepared khaki crystals, the XRD technique was carried out. The XRD patterns of the $TiO_2$ crystals prepared in the presence and absence of shape-controlling agents are shown in Figure 1. The samples were designated as AC-$TiO_2$, AS-$TiO_2$, urea-$TiO_2$, and No-$TiO_2$ according to the addition of shape-controlling agents. The diffraction peaks of AC-$TiO_2$ at $2\theta$ values of 27.50°, 36.20°, 39.30°, 41.32°, 44.14°, 54.42°, 56.68°, 62.84°, 64.21°, 69.16°, and 69.88° were indexed to the (110), (101), (200), (111), (210), (211), (220), (002), (310), (301), and (112) planes of rutile $TiO_2$ (Joint Committee on Powder Diffraction Standards, JCPDS no. 21-1276), while the diffraction peaks at $2\theta$ values of 25.46°, 25.76°, and 30.88° were indexed to the (120), (111), and (121) planes of brookite $TiO_2$ (JCPDS no. 29-1360) as shown in Figure 1a. The percentage of rutile $TiO_2$ (78.3%) and brookite $TiO_2$ (21.7%) in the AC-$TiO_2$ crystals can be estimated using the following equations [16]:

$$W_R = \frac{I_R}{I_R + 2.721 I_B} \quad W_B = \frac{2.721 I_B}{I_R + 2.721 I_B}$$

where $W_R$ and $W_B$ represent the weight fraction of rutile and brookite $TiO_2$, respectively [23,27]. $I_R$ and $I_B$ are the integrated intensity of the rutile $TiO_2$ (110) peak (100.0%) and brookite $TiO_2$ (121) peak (10.2%), respectively [14,16]. For AS-$TiO_2$, the diffraction peaks at $2\theta$ values of 25.34°, 37.08°, 37.88°, 38.64°, 48.10°, 54.00°, 55.14°, 62.86°, 68.88°, 70.30°, and 75.20°, corresponding to the (101), (103), (004), (112), (200), (105), (211), (204), (116), (220), and (215) crystal planes of anatase $TiO_2$ (JCPDS no. 21-1272) as shown in Figure 1b. The diffraction peaks of the urea-$TiO_2$ and No-$TiO_2$ crystals prepared under different conditions corresponded to those of a pure rutile $TiO_2$ (JCPDS no. 21-1276), regardless of the difference in intensity as shown in Figure 1c,d. The diffraction peak intensity of No-$TiO_2$ was stronger than that of urea-$TiO_2$, indicating an improvement in the crystalline sizes and crystallinity.

Figure 2 presents the typical FESEM images of the samples obtained by the solvothermal synthesis method at 160 °C by varying the shape-controlling agents (i.e., AC, AS, and urea) using titanium trichloride ($TiCl_3$) as a precursor and anhydrous and distilled water as solvents. As shown in Figure 2a–c, the AC-$TiO_2$ sample was mainly composed of defined tetragonal-shaped nanorods with a length of 0.15~0.70 μm and width of 15~100 nm and some irregular nanoparticles. Enlarged images revealed that the tetragonal-shaped nanorods were formed by the coalescence of a small number of nanorods. Many tufted microflowers self-assembled by tetragonal-shaped nanorods were observed, indicating that the nanorods began to grow out of the flower clusters [4]. Figure 2d shows the FESEM images of AS-$TiO_2$ microspheres with a diameter of 2.65~5.57 μm. Figure 2e displays an enlarged image of a microsphere, which was self-assembled by nanorods, with a length of several micrometers and width of 20~100 nm along with any directions. As shown in Figure 2f,g, the urea-$TiO_2$ sample was mainly composed of fan-shaped particles, tufted microflowers, and microspheres with an average diameter of 3.50 μm, which were self-assembled by tetragonal-shaped nanorods several micrometers in length and 17~70 nm in width along different directions. Figure 2h,i show the FESEM images of the No-$TiO_2$ sample. It can be seen that the tufted No-$TiO_2$ microflowers were formed by self-assembly of tetragonal-shaped nanorods with a length of 0.30~1.20 μm and a width of 40~180 nm.

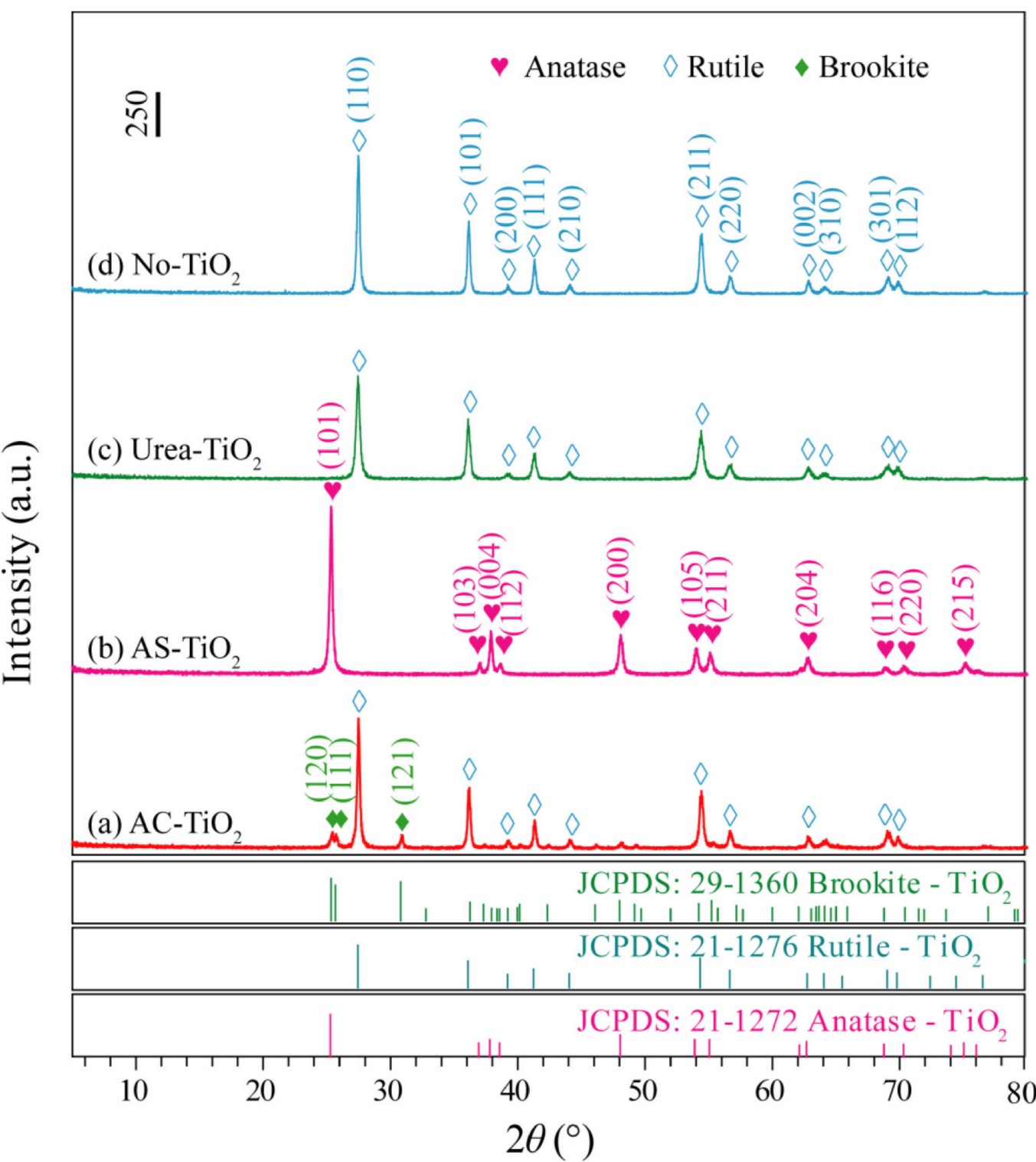

**Figure 1.** XRD patterns of as-prepared (**a**) AC-TiO$_2$; (**b**) AS-TiO$_2$; (**c**) urea-TiO$_2$; (**d**) No-TiO$_2$ micro-crystals in the presence and absence of shape-controlling agents.

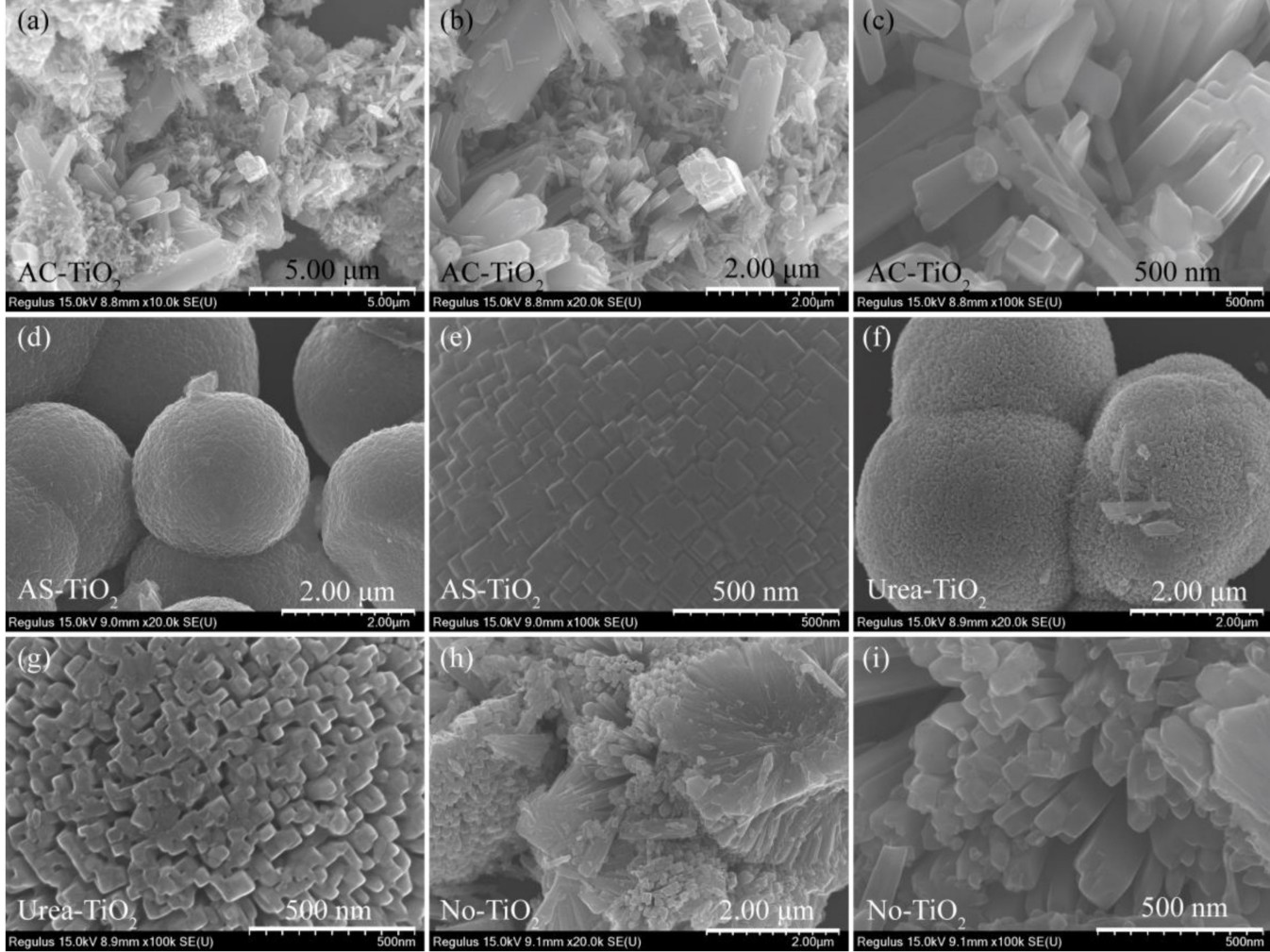

**Figure 2.** Different magnifications of FESEM images of the (**a–c**) AC-TiO$_2$; (**d,e**) AS-TiO$_2$; (**f,g**) urea-TiO$_2$; (**h,i**) No-TiO$_2$ samples in the presence and absence of shape-controlling agents.

The energy-dispersive X-ray spectroscopy (EDS) parameters of prepared TiO$_2$ samples are listed in Table 1. The ready TiO$_2$ samples contained not only titanium and oxygen elements but also a small amount of chlorine (from TiCl$_3$) and other elements from the raw material as shown in Table 1. A tiny amount of carbon in the obtained AC-TiO$_2$ and urea-TiO$_2$ samples came from ammonium carbonate and urea, respectively, and a small amount of sulfur in the obtained AS-TiO$_2$ came from ammonium sulfur.

**Table 1.** EDS analysis of the as-prepared TiO$_2$ samples.

| Sample | AC-TiO$_2$ | | Urea-TiO$_2$ | | AS-TiO$_2$ | | No-TiO$_2$ | |
|---|---|---|---|---|---|---|---|---|
| Element | Atom% | Mass% | Atom% | Mass% | Atom% | Mass% | Atom% | Mass% |
| C | 4.9 | 1.9 | 4.8 | 1.7 | 0.0 | 0.0 | 0.0 | 0.0 |
| N | 0.0 | 0.0 | 0.0 | 0.0 | 0.5 | 0.2 | 0.0 | 0.0 |
| O | 44.6 | 22.4 | 40.9 | 19.8 | 38.0 | 17.2 | 39.6 | 18.0 |
| Cl | 0.2 | 0.2 | 0.4 | 0.4 | 0.1 | 0.1 | 0.2 | 0.2 |
| Ti | 50.3 | 75.5 | 53.9 | 78.1 | 59.9 | 81.2 | 60.2 | 81.8 |
| S | 0.0 | 0.0 | 0.0 | 0.0 | 1.5 | 1.3 | 0.0 | 0.0 |

The facet feature and the crystal growth behavior of the obtained $TiO_2$ samples were further characterized by TEM and HRTEM. Figure 3 shows the TEM and HRTEM images of the AC-$TiO_2$, AS-$TiO_2$, urea-$TiO_2$, and No-$TiO_2$ samples synthesized by the solvothermal process at 160 °C by varying the shape-controlling agents. The TEM image of the AC-$TiO_2$ sample obtained shows a nanorod-shaped morphology with a length of 110~300 nm and a width of 15~45 nm, which is consistent with those observed by FESEM (Figure 3a). For the HRTEM image of AC-$TiO_2$ (Figure 3b), the lattice fringes parallel to the lateral planes were measured to be 0.33 ± 0.01 nm, corresponding to the interplanar distance of the rutile $TiO_2$ (110) planes, indicating that the nanorod are likely to grow along the [001] direction [4]. The fast Fourier transform (FFT) diffraction pattern of the white, dashed lines region (Figure 3b inset) further indicated that the nanorod-shaped $TiO_2$ crystal was single-crystalline. The distances between the lattice fringes, 0.35 ± 0.01 and 0.35 ± 0.01 nm, can be assigned to the (120) and (−120) planes of the brookite $TiO_2$ phase, respectively, and the angle labeled at 80° is identical to the theoretical value for the angle between the (120) and (−120) planes, suggesting that the irregular nanoparticle expose {001} facets on its surface (Figure 3b). The 0.35 ± 0.01 nm of the interplanar spacing of the cuboid-like nanoparticles can be assigned to the (120) planes of the brookite $TiO_2$ (Figure 3c). The above analysis further confirms that the synthesized AC-$TiO_2$ was a mix-phase of rutile and brookite $TiO_2$. Figure 3d–f show TEM and HRTEM images of the AS-$TiO_2$ prepared by the solvothermal treatment of the $TiCl_3$ solution at 160 °C. Dimer particles formed by the aggregation of many nanorods in a specific direction were observed as shown in Figure 3d. The visible lattice fringes with an interplanar spacing of 0.35 ± 0.01 nm matched well with the (101) crystal plane of tetragonal anatase $TiO_2$ (Figure 3e). Viewed along the [010] direction, there were two atomic planes (101) and (002), the lattice spacing was 0.35 ± 0.01 and 0.48 ± 0.01 nm, respectively, and the interface angle was 68.3°, further indicating that the as-prepared AS-$TiO_2$ sample was a single-crystalline anatase $TiO_2$ (Figure 3f) [28,29]. Figure 3g–i present the TEM and HRTEM images of the urea-$TiO_2$ sample obtained by solvothermal reaction of the $TiCl_3$ solution with urea as the shape-controlling agent at 160 °C. Three sets of lattice fringes with intervals of 0.25 ± 0.01, 0.25 ± 0.01, and 0.33 ± 0.01 nm and angles of 45°, 67.5°, and 67.5° can be identified in the HRTEM image, which is in good agreement with the spacing of the (101), (011), and (110) planes of the tetragonal rutile $TiO_2$, respectively, indicating that the as-prepared urea-$TiO_2$ nanorods were single-crystalline. Figure 3i shows a lattice image from the top and side of a tetragonal-shaped nanorod. The distance between two consecutive planes was measured to be 0.33 ± 0.01 nm, which matches well with the distance between the (110) planes of tetragonal rutile $TiO_2$. Furthermore, the lattice fringes were parallel to the lateral planes, indicating that the tetragonal rutile $TiO_2$ grew along the [001] direction. The FFT diffraction pattern of the white, dashed line region (Figure 3i inset) further indicates that the tetragonal rutile $TiO_2$ crystal was a single-crystalline and grew along the [001] direction. Figure 3j shows a TEM image of the tetragonal-shaped No-$TiO_2$ nanorods obtained by solvothermal reaction at 160 °C for 24 h in the absence of a shape-controlling agent. Figure 3k shows a typical HRTEM image of a tetragonal-shaped No-$TiO_2$ nanorod. The growth direction of the No-$TiO_2$ nanorod can be determined as [001] direction by the FFT diffraction pattern (Figure 3k inset). Figure 3l presents a lattice image taken from the middle of a nanorod. The existence of the two atomic planes, (110) and (001), with a lattice spacing of 0.33 ± 0.01 and 0.30 ± 0.01 nm between two consecutive planes and an interfacial angle of 90° between the (110) and (001) crystal planes, proves that the tetragonal-shaped No-$TiO_2$ nanorod should be mainly exposed with {001} facets on its top planes, and grow along the [001] direction. According to the above structural analysis, the tetragonal-shaped rutile $TiO_2$ nanorods (i.e., AC-$TiO_2$, urea-$TiO_2$, and No-$TiO_2$) were mainly exposed to {001} facets on their top planes, and grew along the [001] direction, while the exposed crystal facets of the AS-$TiO_2$ anatase nanorods on the basal surface is uncertain.

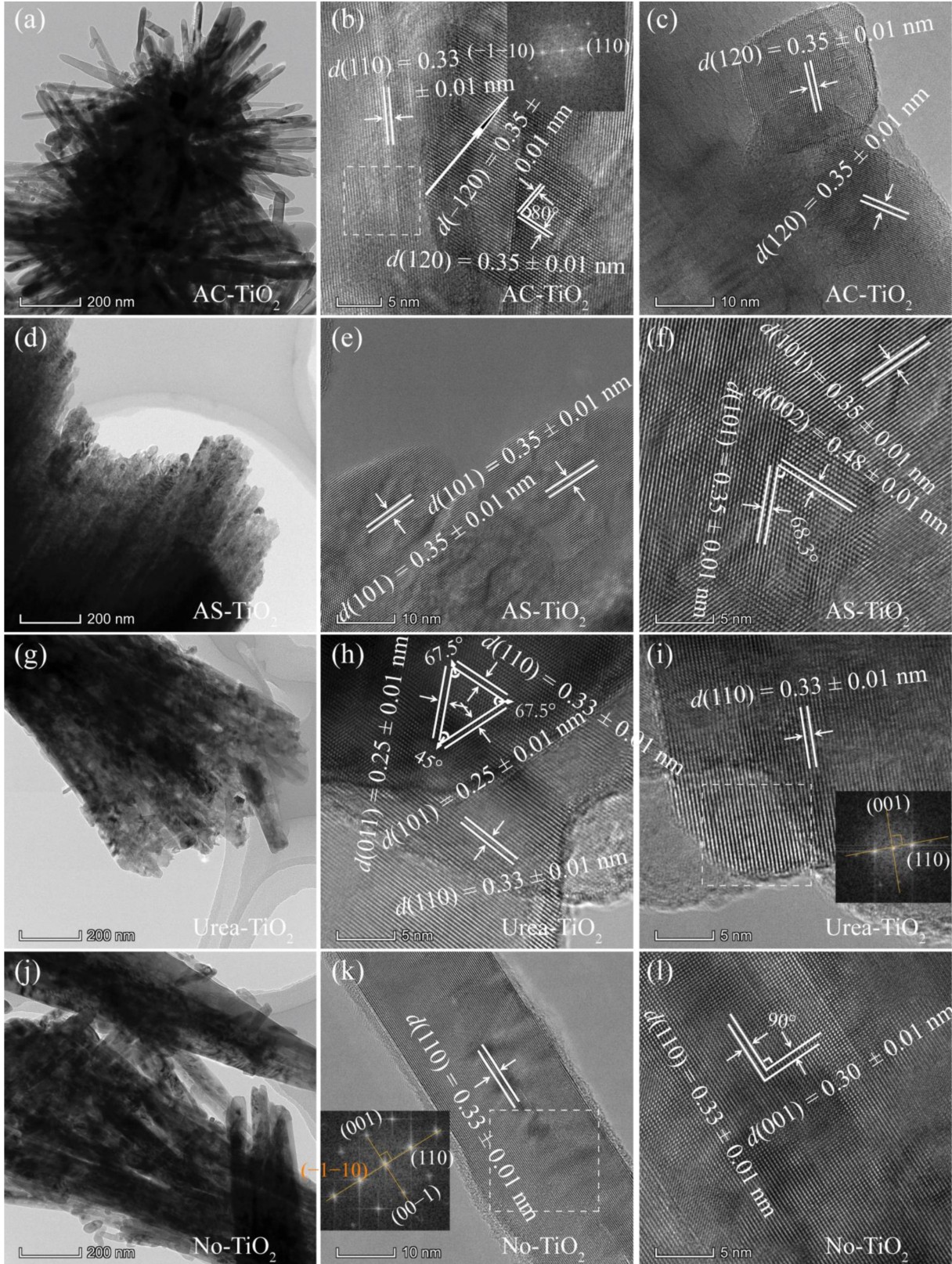

**Figure 3.** TEM and HRTEM images of the prepared (**a**–**c**) AC-TiO$_2$; (**d**–**f**) AS-TiO$_2$; (**g**–**i**) urea-TiO$_2$; (**j**–**l**) No-TiO$_2$ samples in the presence and absence of shape-controlling agents. The insets in (**b**,**i**,**k**) are fast Fourier transform (FFT) diffraction patterns.

### 2.2. X-ray Photoelectron Spectroscopy Analysis

The surface chemical states of the pure anatase, rutile, and mixed-phase $TiO_2$ were studied with X-ray photoelectron spectroscopy (XPS). Figure 4 shows the surveys of the AC-$TiO_2$, AS-$TiO_2$, urea-$TiO_2$, No-$TiO_2$, and CM-$TiO_2$. Only the correlation peaks of C, Ti, and O are observed in Figure 4a, indicating that the chemical purities of the synthetic and commercial samples were very high. Figure 4b displays the Ti 2p spectra of AC-$TiO_2$, AS-$TiO_2$, urea-$TiO_2$, No-$TiO_2$, and CM-$TiO_2$. Two typical peaks at binding energies of 458.6 and 464.4 eV (or 458.9 and 464.8 eV) observed in the as-prepared and commercial $TiO_2$ samples can be attributed to the $Ti^{4+}$ states of $TiO_2$ [30]. As for the AC-$TiO_2$, AS-$TiO_2$, urea-$TiO_2$, No-$TiO_2$s, and CM-$TiO_2$ samples, the corresponding binding energies were centered at 458.6 (Ti $2p_{2/3}$) and 464.4 eV (Ti $2p_{1/2}$) for the AC-$TiO_2$, urea-$TiO_2$, No-$TiO_2$, and CM-$TiO_2$ samples, while the binding energies were centered at 458.9 (Ti $2p_{2/3}$) and 464.8 eV (Ti $2p_{1/2}$) for the AS-$TiO_2$ sample. This slight discrepancy in binding energies can be attributed to the different surface atomic arrangements and configurations of the as-prepared $TiO_2$ and CM-$TiO_2$ samples with different crystal phases [30,31]. The Ti $2p_{2/3}$ peaks shifted from 458.9 of the AS-$TiO_2$ to 458.6 eV for the AC-$TiO_2$, urea-$TiO_2$, No-$TiO_2$, and CM-$TiO_2$ accompanying the negative shift of the Ti $2p_{1/2}$ peaks from 464.8 to 464.4 eV, suggesting the partial reduction of $TiO_2$ with the formation of $Ti^{3+}$ ions on the surface of the as-prepared AC-$TiO_2$, urea-$TiO_2$, No-$TiO_2$, and CM-$TiO_2$ [32]. The high-resolution O 1s and C 1s XPS spectra of the AC-$TiO_2$, AS-$TiO_2$, urea-$TiO_2$, No-$TiO_2$, and CM-$TiO_2$ are shown in Figure 4c,d, respectively. The singe peak for O 1s at 529.8 eV (or 530.2 eV) corresponded to the crystal lattice oxygen in the as-prepared and the commercial $TiO_2$, and the O 1s peaks shifted from 530.2 for the AS-$TiO_2$ to 529.8 eV for the AC-$TiO_2$, urea-$TiO_2$, No-$TiO_2$, and CM-$TiO_2$, suggesting the existence of more oxygen vacancies on the surface of the as-prepared AC-$TiO_2$, urea-$TiO_2$, No-$TiO_2$, and CM-$TiO_2$ [32,33]. Furthermore, the khaki colors of the as-prepared $TiO_2$ samples further confirmed the existence of $Ti^{3+}$ ions and oxygen vacancies [30] as shown by their digital photograph presented in Figure 8. The two peaks for C 1s at 284.8 and 288.6 eV corresponded to the C–O and C=O bonds in hydrocarbons [34]. In addition, signals related to N were detected at 400.1 eV in the spectra of AC-$TiO_2$, AS-$TiO_2$, and urea-$TiO_2$ (Figure 4e), which was assigned to N 1s from the imino group (=NH) of the ammonium carbonate, ammonium sulfate, and urea, respectively. Signals related to S were detected at 168.9 eV in the spectra of AS-$TiO_2$, which was assigned to S $2p_{1/2}$ from the thiol group (–SH) of ammonium sulfate (Figure 4f).

### 2.3. Photoluminescence Analysis

To understand the origin of the different photocatalytic activities of the prepared AC-$TiO_2$, AS-$TiO_2$, urea-$TiO_2$, and No-$TiO_2$ samples, the recombination of photogenerated electron-hole pairs was studied by PL spectroscopy. Figure 5 exhibits the photoluminescence (PL) spectra of the prepared AC-$TiO_2$, AS-$TiO_2$, urea-$TiO_2$, and No-$TiO_2$ samples excited by 325 nm. As reported, a stronger intensity of PL indicates higher efficiency of electron and hole recombination and, as a consequence, worse charge separation [35]. However, PL emission intensity can be affected by many other factors such as the particle size (PL emission intensity increases with the decrease in particle size), dopant species, semiconductor nanomaterial form, geometry of the particles (which can affect the light diffusion), and number of active defects [36]. The observed broad visible photoluminescence was mainly related to self-trapped excitons and oxygen vacancy-related defect states in the as-prepared $TiO_2$ samples [9]. As shown in Figure 5, the pronounced emission peaks at 558 nm were assigned to oxygen vacancies in the AS-$TiO_2$, AC-$TiO_2$, No-$TiO_2$, and urea-$TiO_2$ samples, which were caused by the rapid recombination of photogenerated electron-hole pairs [37,38]. Except for AS-$TiO_2$, the PL emission spectra of all $TiO_2$ located between 350 and 650 nm, and the emission peak at 394 nm can be attributed to the emission of the band-band PL process of $TiO_2$; the other emission peaks at 394, 436, 467, 481, and 616 nm can be attributed to the excitonic PL process at the band edge of $TiO_2$ [36,39]. In particular, the emission peaks of the as-prepared $TiO_2$ at 467 and 616 nm can be attributed

to the shallow trap states originated from oxygen vacancies associated with $Ti^{3+}$ ions and the intrinsic defect, respectively [40]. The intensity of PL for the $TiO_2$ crystals gradually increased following the order of AS-$TiO_2$, AC-$TiO_2$, No-$TiO_2$, and urea-$TiO_2$, demonstrating that the recombination rate of photogenerated electron-hole pairs increased gradually. The AS-$TiO_2$ crystals had the most robust separation efficiency and the lowest recombination rate, which is conducive to the improvement in the photocatalytic performance.

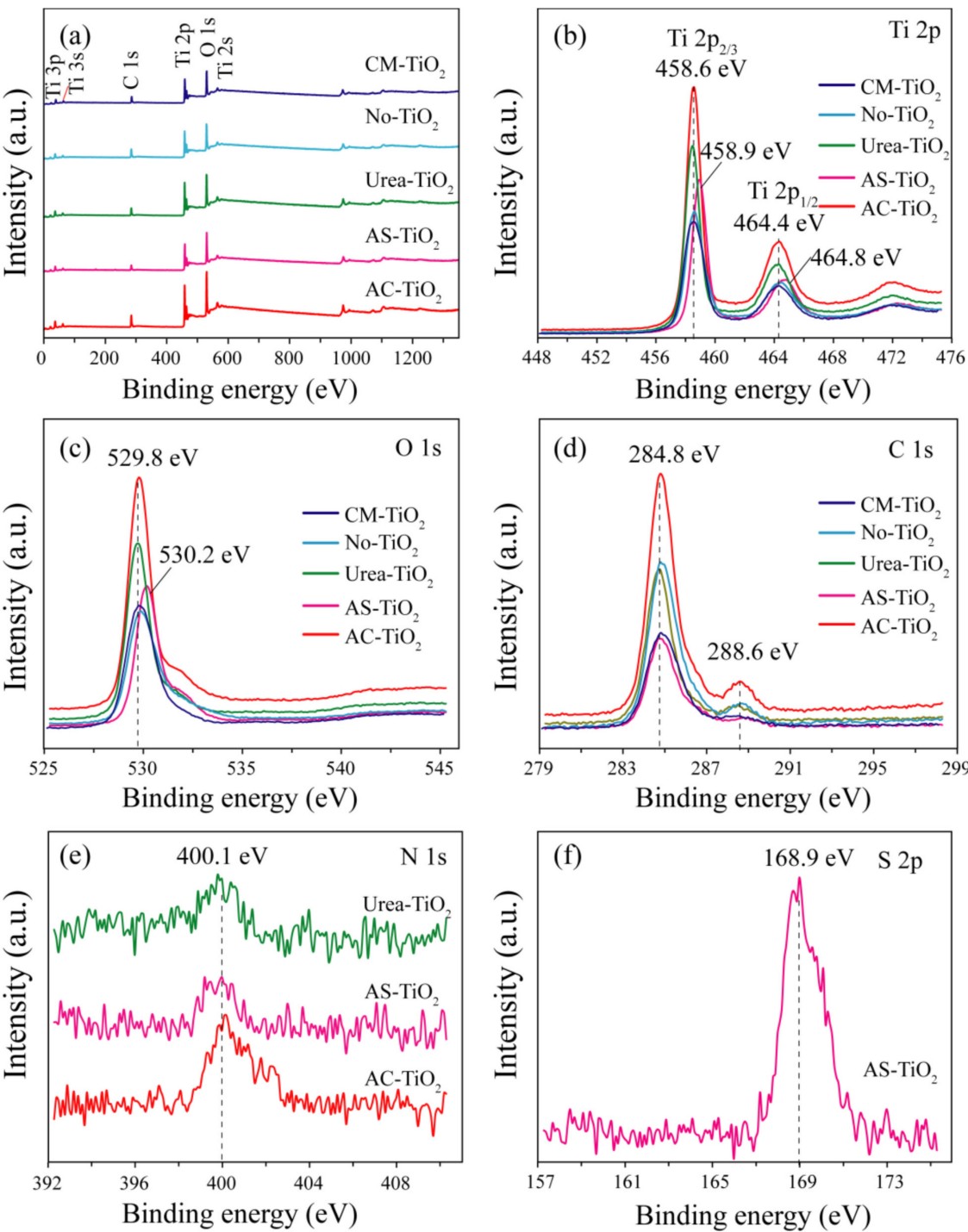

**Figure 4.** XPS spectra of AC-$TiO_2$, AS-$TiO_2$, urea-$TiO_2$, No-$TiO_2$, and CM-$TiO_2$: (**a**) survey spectra; (**b**) Ti 2p spectra; (**c**) O 1s spectra; (**d**) C 1s spectra; (**e**) N 1s spectra; (**f**) S 2p spectra.

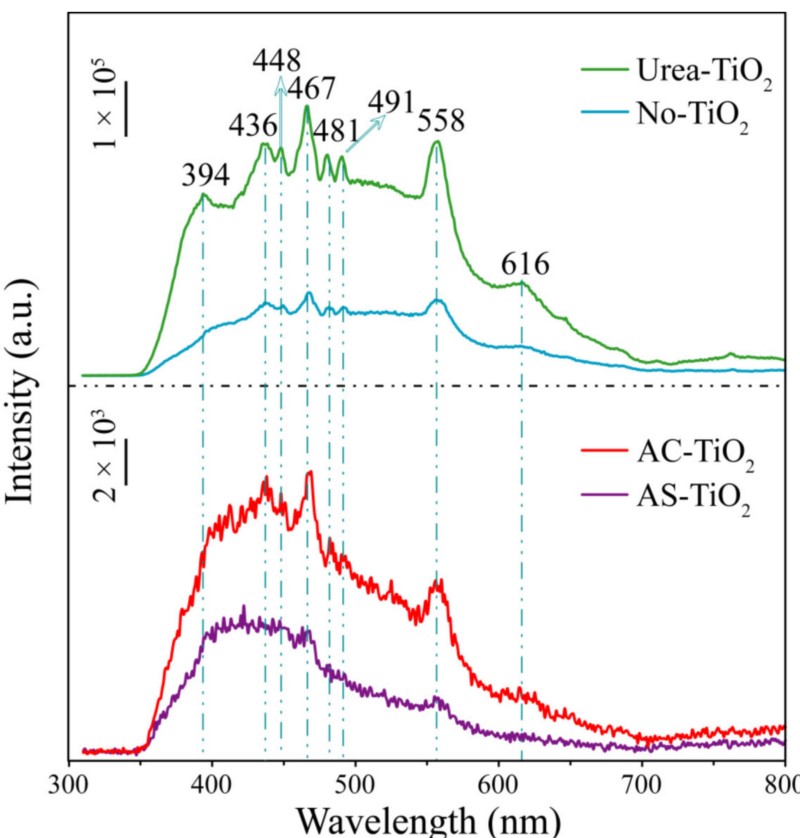

**Figure 5.** PL emission spectra (excited at 325 nm) of the prepared AC-TiO$_2$, AS-TiO$_2$, urea-TiO$_2$, and No-TiO$_2$ samples.

### 2.4. Electrochemical Impedance Spectroscopy Analysis

The electron-transfer rate of the as-prepared TiO$_2$ samples was investigated by electrochemical impedance spectroscopy (EIS) as shown in Figure 6. It can be seen that the semicircle radius of the AC-TiO$_2$ sample was smaller than that of AS-TiO$_2$, urea-TiO$_2$, and No-TiO$_2$ in the high-frequency region of the EIS Nyquist plot. In the EIS Nyquist plot, the semicircle radius was related to the electrode resistance, which decreased with electrode resistance. It is universally acknowledged that the separation efficiency of photogenerated electron-hole pairs has an essential impact on the photocatalytic activity. The high conductivity of the AS-TiO$_2$ sample also facilitates electron transfer, resulting in an efficient charge separation [41]. The low conductivity of the AC-TiO$_2$, urea-TiO$_2$, and No-TiO$_2$ samples with the same trend was not conducive to electron transfer, resulting in an inefficient electron-hole separation.

### 2.5. Photocatalytic Activity

The photocatalytic activity of AC-TiO$_2$, AS-TiO$_2$, urea-TiO$_2$, and No-TiO$_2$ was evaluated by the degradation of organic dye MB under UV irradiation. To better evaluate their photocatalytic efficiency, CM-TiO$_2$ was selected as the photocatalytic benchmark. Figure 7a shows the temporary changes in the adsorption spectra of MB with an irradiation time in the presence of AS-TiO$_2$. The intensity and wavelength of the maximal absorption peak in the visible region gradually decreased and shifted from 664 to 649 nm with increasing UV irradiation time, respectively, indicating the degradation and *N*-demethylation of MB [42]. Figure 7b displays the variation in the MB photocatalytic degradation efficiency with the irradiation time in the presence and absence of TiO$_2$ samples. In the absence of TiO$_2$, the self-degradation efficiency of MB in the blank test was low, only 6.5% at 120 min. In the presence of TiO$_2$, the degradation efficiency of MB increased on the order of 33.6% (No-TiO$_2$), 35.5% (urea-TiO$_2$), 38.1% (AC-TiO$_2$), 59.7% (CM-TiO$_2$), and 70.4% (AS-TiO$_2$) at 120 min. AS-TiO$_2$

exhibited enhanced photocatalytic activity under UV light irradiation, increasing by a factor of 10.83, 2.09, 1.98, 1.85, and 1.18 compared with that of the blank, No-TiO$_2$, urea-TiO$_2$, AC-TiO$_2$, and CM-TiO$_2$, respectively. Many factors, such as the crystalline phase, heterojunction structure, morphology, crystallinity, crystallite size, specific surface area, exposed facets, the charge carrier's generation, and photocatalytic performance, have an important influence on photocatalytic activity [36,43]. The specific surface area increased in the order of No-TiO$_2$ (2.58 m$^2$/g), urea-TiO$_2$ (4.71 m$^2$/g), CM-TiO$_2$ (7.27 m$^2$/g), AS-TiO$_2$ (9.93 m$^2$/g), and AC-TiO$_2$ (12.82 m$^2$/g), which is not completely consistent with the order of the photocatalytic activity, indicating that there are other factors affecting the photocatalytic activity. Among the four crystalline phases of TiO$_2$, anatase usually has the highest photocatalytic activity for the degradation of organic dye molecules [44]. Among the five TiO$_2$ samples, AS-TiO$_2$ showed the highest photocatalytic activity, which can be attributed to its anatase phase structure and relatively high specific surface area. The relatively high photocatalytic activity of CM-TiO$_2$ can be attributed to the different energy band structures of anatase (96.8%) and rutile (3.2%), because the photogenerated electrons transfer from rutile to anatase, whereas the photogenerated holes transfer from anatase to rutile, which can inhibit the charge recombination of the photogenerated electron-hole pairs, thereby enhancing the photocatalytic activity [45,46]. No-TiO$_2$ and urea-TiO$_2$ delivered the lower photocatalytic activity with a degradation efficiency of 38.1% and 35.5%, respectively, because these two samples consisted only of single rutile and had smaller specific surface areas (No-TiO$_2$: 2.58 m$^2$/g; urea-TiO$_2$: 4.71 m$^2$/g). Moreover, the rutile had a direct band gap structure, resulting in the rapid recombination of photogenerated electron-hole pairs [47]. Despite the largest surface area for AC-TiO$_2$ (rutile: 78.3%; brookite: 21.7%) compared to the other samples, lower photocatalytic activity is observed, with a degradation efficiency of 38.1%. This was because rutile and brookite TiO$_2$ have direct band gap structures that cannot provide longer electron-hole lifetimes, leading to the fast recombination of photogenerated electron-hole pairs [47,48]. To characterize the surface photocatalytic activity of the as-prepared TiO$_2$ crystals, the photodegradation amount of MB per surface area was estimated according to the formula, degradation amount per surface area (mg(MB)/m$^2$(TiO$_2$)) = $(m_0(MB) - m_t(MB))/(m(TiO_2) \cdot S_{BET}(TiO_2))$, where $m_0(MB)$ and $m_t(MB)$ represent the quality of MB in solution before and after illumination for a certain time, respectively; $m(TiO_2)$ and $S_{BET}(TiO_2)$ are the quality and specific surface area of TiO$_2$, respectively [49]. The photodegradation amount of MB at 120 min was 1.59, 0.88, 0.85, 0.84, and 0.32 mg (MB)/m$^2$(TiO$_2$) for No-TiO$_2$, urea-TiO$_2$, CM-TiO$_2$, AS-TiO$_2$, and AC-TiO$_2$, respectively, as shown in Figure 7c. The maximum amount of photodegradation per specific surface area of No-TiO$_2$ (1.59 mg(MB)/m$^2$(TiO$_2$)) was due to the small differences in the amount of degradation per gram of MB (4.10, 4.13, 6.21, 8.37, and 4.14 mg(MB)/g(TiO$_2$) for No-TiO$_2$, urea-TiO$_2$, CM-TiO$_2$, AS-TiO$_2$, and AC-TiO$_2$, respectively), while its specific surface area was far lower than that of other TiO$_2$ samples (2.58, 4.71, 7.27, 9.93, and 12.82 m$^2$/g for No-TiO$_2$, urea-TiO$_2$, CM-TiO$_2$, AS-TiO$_2$, and AC-TiO$_2$, respectively). The degradation amount of urea-TiO$_2$ (4.13/4.71 = 0.88), CM-TiO$_2$ (6.21/7.27 = 0.85), and AS-TiO$_2$ (8.37/9.93 = 0.84) per specific surface area was almost the same, although the degradation amount per gram (4.13, 6.21, and 8.37 mg(MB)/g(TiO$_2$) for urea-TiO$_2$, CM-TiO$_2$, and AS-TiO$_2$, respectively) and specific surface area (4.71, 7.27, and 9.93 m$^2$/g for urea-TiO$_2$, CM-TiO$_2$, and AS-TiO$_2$, respectively) were different, which may be due to the presence of rutile in CM-TiO$_2$ and specific crystal planes in urea-TiO$_2$ and AS-TiO$_2$. The smallest photodegradation amount of AC-TiO$_2$ (0.32 mg(MB)/m$^2$(TiO$_2$)) per specific surface area was due to the smaller degradation amount per gram (4.14 mg(MB)/g(TiO$_2$)) and the largest specific surface area (12.82 m$^2$/g). The stability and reusability of the AS-TiO$_2$ and CM-TiO$_2$ were evaluated by carrying out cycling experiments three times for the photodegradation of MB as shown in Figure 7d. The degradation rates of the AS-TiO$_2$ and CM-TiO$_2$ samples decreased by only 5.89% and 6.35% for the MB photodegradation, respectively, after recycling three times, indicating the AS-TiO$_2$ and CM-TiO$_2$ samples possessed good stability and reusability.

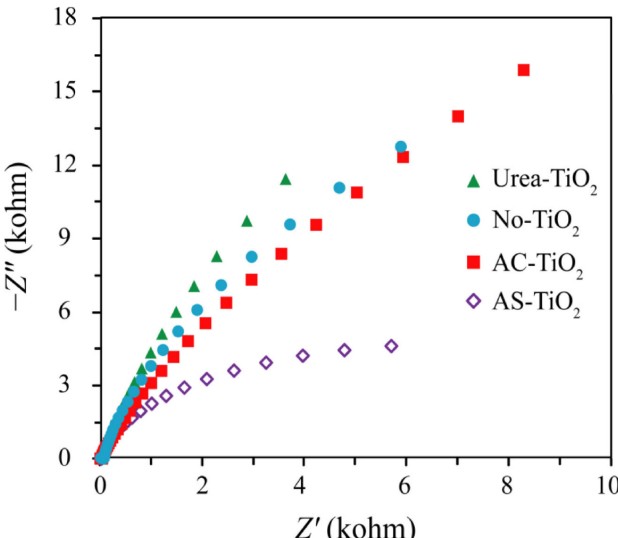

**Figure 6.** Electrochemical impedance spectroscopy Nyquist plots of AC-TiO$_2$, AS-TiO$_2$, urea-TiO$_2$, and No-TiO$_2$ electrodes in 0.2 M Na$_2$SO$_4$ aqueous solution.

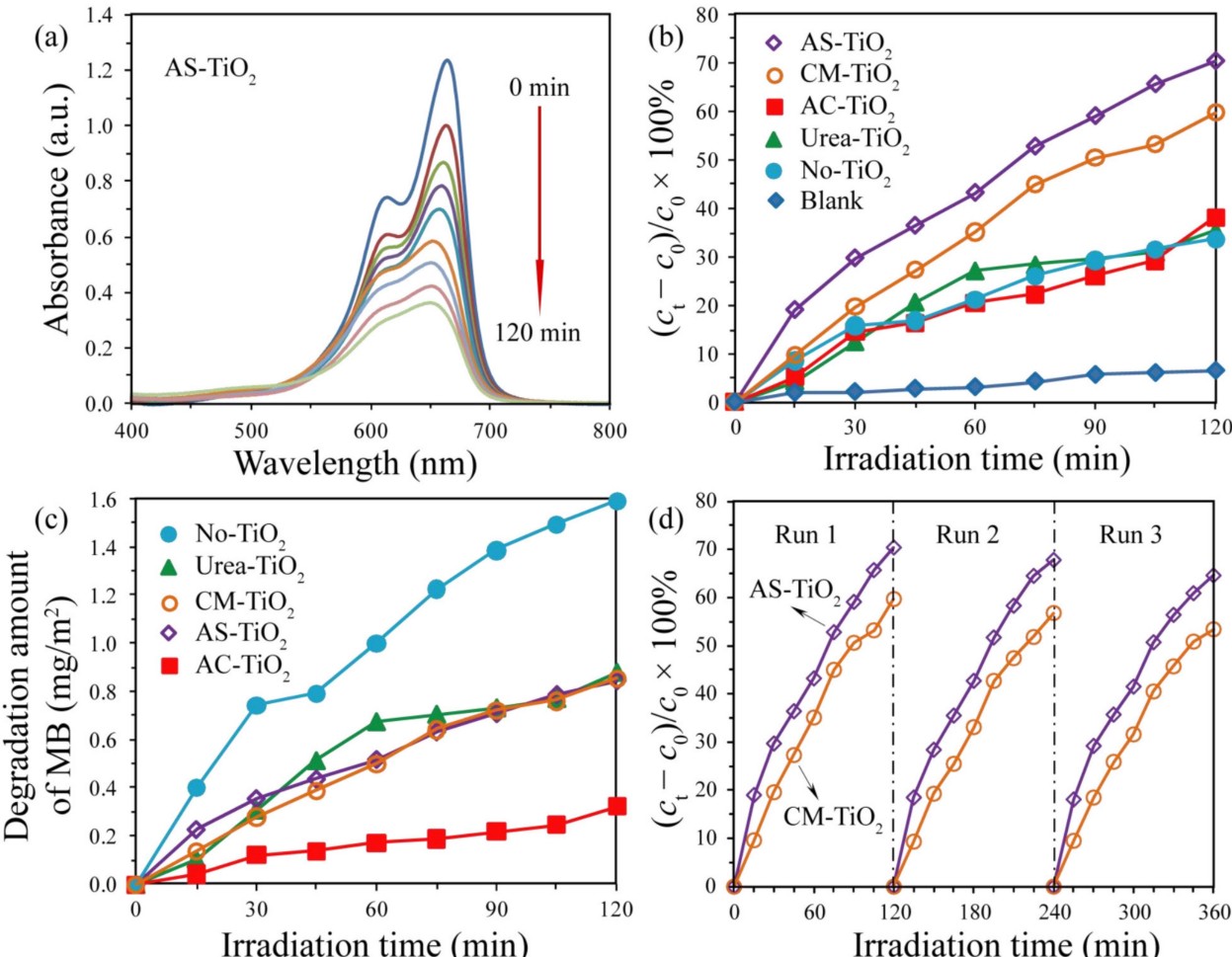

**Figure 7.** (**a**) Temporal adsorption spectral changes of MB in aqueous AS-TiO$_2$ suspensions under UV illumination; (**b**) photocatalysis degradation efficiency of MB with or without TiO$_2$ sample under UV illumination; (**c**) photocatalysis degradation amount of MB with different TiO$_2$ samples under UV illumination; (**d**) recycled performances in the presence of AS-TiO$_2$ and CM-TiO$_2$ samples for photodegradation of MB dye.

## 3. Materials and Methods

### 3.1. Materials

Titanium trichloride (TiCl$_3$, 15~20%) and ammonium sulfate ((NH$_4$)$_2$SO$_4$, 99.0%) were purchased from Tianjin Beichen District Fangzheng Reagent Factory (Tianjin, China). Ammonium carbonate (NH$_4$NH$_2$CO$_2$ + NH$_4$HCO$_3$), absolute ethyl alcohol (C$_2$H$_5$OH, 99.7%), and urea were purchased from Tianjin Bodi Chemical Co., Ltd., Tianjin Kemiou Chemical Reagent Co., Ltd. (Tianjin, China), and Shanghai Macklin Biochemical Technology Co., Ltd. (Shanghai, China), respectively. All the chemical reagents used in the experiments were not further purified.

### 3.2. Synthesis of TiO$_2$ Nanocrystals

Firstly, 100 mL of absolute ethyl alcohol and 100 mL deionized water were poured into a round-bottomed flask with a capacity of 500 mL. Then, 50 mL of TiCl$_3$ was added dropwise to the above round-bottomed flask under magnetic stirring. Thirty minutes later, 62.5 mL of the above purple mixed solution was transformed into three Teflon-lined stainless-steel autoclaves with a capacity of 80 mL. Then, 2.5001 g of ammonium sulfate (AS), 2.5013 g of ammonium carbonate (AC), and 2.5014 g of urea were added to the above autoclaves, respectively. For comparison, 62.5 mL of the above purple solution was transformed into the same autoclave without adding any other chemical reagents. After, the above four autoclaves were placed in a constant temperature blast drying oven and heated at 160 °C for 24 h. The khaki powders obtained were denoted as AS-TiO$_2$, AC-TiO$_2$, urea-TiO$_2$, and No-TiO$_2$, respectively (Figure 8).

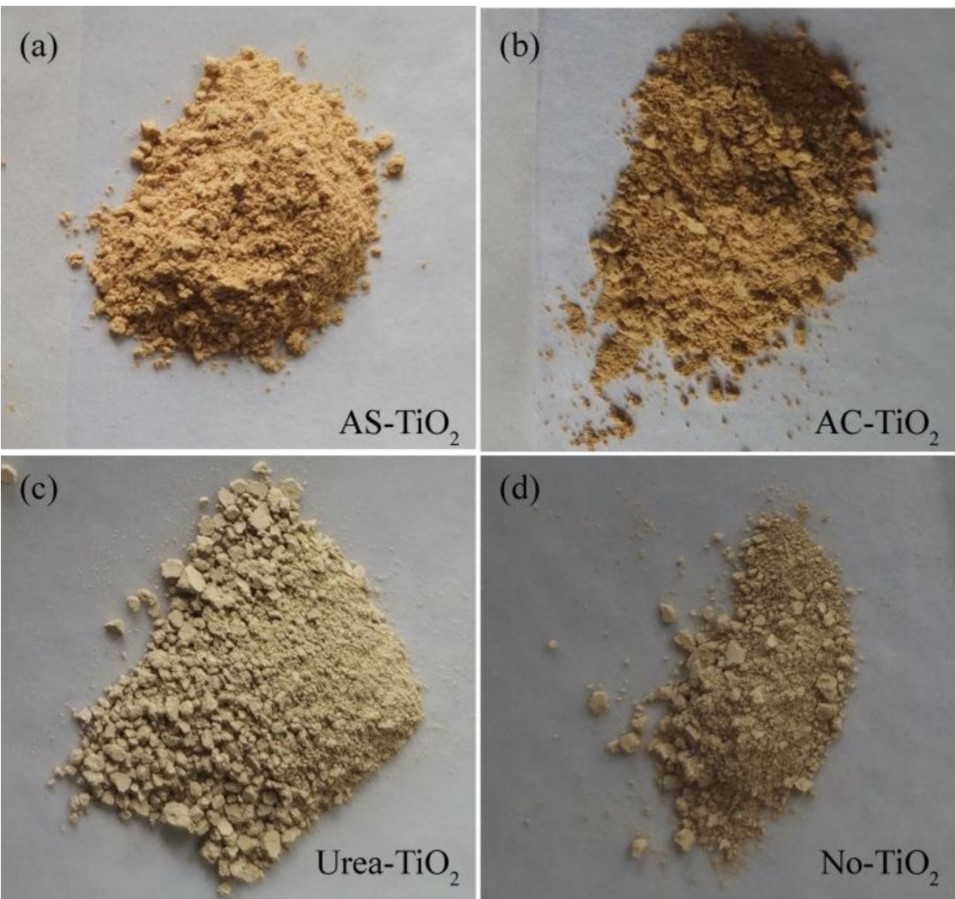

**Figure 8.** Photographs of the as-prepared (**a**) AS-TiO$_2$; (**b**) AC-TiO$_2$; (**c**) urea-TiO$_2$; (**d**) No-TiO$_2$ microcrystals in the presence and absence of shape-controlling agents.

### 3.3. Sample Characterization

The crystallite structure of the khaki $TiO_2$ nanocrystals was determined by powder X-ray diffraction (XRD-6100, Shimadzu, Kyoto, Japan) with monochromated Cu K$\alpha$ radiation ($\lambda$ = 0.15406 nm) at a scan speed of 8°/min, an accelerating voltage of 40 kV and applied current of 30 mA. The morphology and microstructure of the samples were examined with a field emission scanning electron microscope (FESEM, Hitachi SU8100, Tokyo, Japan) equipped with energy-dispersive X-ray spectroscopy (EDS) at an accelerating voltage of 15 kV and an applied current of 10 µA in the darkfield mode. After the sample was drop-casted on silicon wafers, a transmission electron microscope (TEM) and a high-resolution transmission electron microscope (HRTEM, FEI TALO F200S, Portland, OR, USA) with a 200 kV operating voltage were used after the sample was deposited on a standard copper grid-supported carbon film. The elemental compositions and chemical status were analyzed by X-ray photoelectron spectroscopy (XPS, Thermo Fisher Scientific K-Alpha, New York, NY, USA) fitted with an Al K$\alpha$ source (1486.6 eV). The specific surface area was determined by nitrogen gas adsorption (micromeritics ASAP 2020, Micromeritics Instrument Corp., Atlanta, GA, USA). The optical property, carrier migration, and recombination were studied by a fluorescence spectrometer (PL, HORIBA Fluoromax-4, HORIBA Instruments Inc., Kyoto, Japan) and electrochemical impedance spectroscopy (EIS, CHI600E, Shanghai Chenhua Instrument Co., Ltd., Shanghai, China), respectively. The photoelectrochemical measurement was carried out on the electrochemical workstation under the irradiation of a 300 W xenon lamp equipped with a cut-off filter ($\lambda$ > 420 nm). TFO conductive glass (opening area: 1 $cm^2$), platinum, and an Ag/AgCl electrode were used as the working electrode, the counter electrode, and the reference electrode, respectively, and 0.2 mol/L $Na_2SO_4$ solution was used as electrolyte. EIS was recorded in the frequency range of 100 kHz to 0.01 Hz under open circuit potential conditions. The absorbance characteristics of the MB solution were determined by a UV-Vis spectrophotometer (TU 1901, Beijing Purkinje General Instrument Co., Ltd., Beijing, China).

### 3.4. Photocatalytic Experiments

Photocatalytic activity of the synthesized khaki $TiO_2$ samples was evaluated with methylene blue (MB) as the model pollutant under UV irradiation. Typically, 75 mg of khaki $TiO_2$ sample (AS-$TiO_2$, AC-$TiO_2$, urea-$TiO_2$, and No-$TiO_2$) and 150 mL of 10 mg/L MB ($2.23 \times 10^{-5}$ mol/L) sample were mixed and subjected to magnetic stirring in the dark without irradiation for 2 h to establish the adsorption-desorption equilibrium of the MB dye on the surface of $TiO_2$ samples. A low-pressure mercury lamp irradiation (175 W, $\lambda_{max}$ = 365 nm, Shanghai Mingyao Glass Hardware Tool Factory, Shanghai, China) with a maximum emission at 365 nm was used as the UV resource, and the distance between the mercury lamp and the suspension surface was 25 cm. Within a given irradiation interval, 3 mL of suspension was taken out and analyzed after removing the solid particles by centrifugation. The concentration change of the MB solution was detected by an ultraviolet-visible spectrophotometer. As a comparison, the photocatalytic activity of the commercial $TiO_2$ (CM-$TiO_2$, 96.8% anatase and 3.2% rutile) sample was also measured under the same conditions. As for stability and recyclability, the $TiO_2$ sample was filtered with a sand core filter and thoroughly dried after each cycle, and then a new MB solution was added for further analysis.

### 4. Conclusions

In summary, mixed-phase AC-$TiO_2$ crystals grew along the [001] direction (rutile nanorods) and were exposed to {001} facets (irregular brookite nanoparticles) on their basal surface; anatase AS-$TiO_2$ microspheres were formed by the self-assembly of nanorods with uncertain facets; rutile urea-$TiO_2$ microspheres and tufted rutile No-$TiO_2$ microflowers were formed by the self-assembly of nanorods with oriented growth along the [001] direction, and exposed {001} facets on their top surface were synthesized via a facile solvothermal route in the presence of $TiCl_3$ and different morphology-controlling agents. The AS-$TiO_2$

microspheres exhibited the highest photocatalytic activity towards decoloration of the MB solution and achieved 70.4% degradation for the MB solution in 120 min, almost 10.83, 2.09, 1.98, 1.85, and 1.18 times as high as that of the blank, No-TiO$_2$, urea-TiO$_2$, AC-TiO$_2$, and CM-TiO$_2$, respectively. Based on the results of XRD, FESEM, HRTEM, specific surface area, PL, and EIS, the high photocatalytic activity of the AS-TiO$_2$ microspheres can be attributed to the combined effect of the anatase phase structure, relatively larger specific surface area, the highest separation efficiency, and the lowest recombination rate. This work provides a simple and economical method to synthesize different TiO$_2$ crystals with exposed specific crystal surfaces for the photodegradation of organic pollutants.

**Author Contributions:** Conceptualization, Y.-e.D. and X.N.; methodology, Y.-e.D. and X.N.; formal analysis, K.H. and X.H.; writing—original draft preparation, Y.-e.D.; writing—review and editing, Y.-e.D. and C.Z.; funding acquisition, Y.-e.D. and C.Z. All authors have read and agreed to the published version of the manuscript.

**Funding:** This research was supported by grants 201901D111303 and 201801D121257 from the Applied Basic Research Project of Shanxi; grant 2019L0881 from the Shanxi Scientific and Technological Innovation Programs of Higher Education Institutions; grant PY201817 from the Shanxi "1331 Project" Key Innovation Team; grant I018038 from the Shanxi "1331 Project" Collaborative Innovation Center Fund Project; grant jzxycxtd2019005 from the Jinzhong University "1331 Project" Key Innovation Team; Research Start-up Fee of Jinzhong University.

**Data Availability Statement:** Data are contained within the article and are also available from the first corresponding author.

**Conflicts of Interest:** The authors declare no conflict of interest.

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
