# Peer review of "Microflowery, Microspherical, and Fan-Shaped TiO2 Crystals via Hierarchical Self-Assembly of Nanorods with Exposed Specific Crystal Facets and Enhanced Photocatalytic Performance"

_catalysts, doi:10.3390/catal12020232_

Round 1

Reviewer 1 Report

The authors describe the synthesis of TiO2 of different polymorphs and with varying morphology, using a hydrothermal route based on TiCl3 and various shape-controlling reagents. The products are thoroughly characterized (through XRD, SEM, TEM and XPS) and evaluated for their photocatalytic performance (PL and model reaction of dye degradation).

I am missing a clear statement in the introduction on the novelty and significance of the work. This would help the reader in understanding the purpose of the work, but it is also crucial for me as a reviewer to be able to give a fair evaluation. This must in my opinion be improved in a revised version.

Is the novelty the synthesis method or the resulting morphologies? In that case the authors need to provide more information on how these differ from previous reports. Looking at the review by G. Liu et al. (Chem. Rev. 2014, 114, 9559) I can see superficially similar particles, in terms of structure and morphology, reported before. The chloride-based chemistry is also described in the same review.

Does the novelty lie in deducing the origin of the differences in activity for the different surfaces or polymorphs? In that case I find the series of samples lacking: particle size, morphology, structure, surface area, contaminants etc. all vary between the samples, making it impossible to pinpoint the origin of the changes in activity. Particularly the effect of leftover morphology-controlling agents is important to take into account. As noted in ref. 18, these can mask the underlying effects from structure and morphology. As different contaminants are present on all the samples, it becomes even more difficult to evaluate their reactivity in a systematic way.

This explains my low scores on novelty, significance and interest. With a clearer guide from the authors on how to judge these points, I might have given higher scores, even without changing the content of the rest of the manuscript.

Some additional important points are listed below:

For the TEM analysis of the faceting, it appears as if the authors find sets of lattice fringes in the images, e.g. anatase (101) and (002) in figure 3f and thereby conclude that the crystal has a {010} faceting. This is inappropriate: the presence of (101) and (002) fringes indicates a [010] viewing direction, but says nothing about the facets present on the top/bottom of the crystal; these top/bottom facets need not be perpendicular to the viewing direction. To deduce faceting from TEM, you can look at the planes parallel to the viewing direction at the perimeter of the crystal, ideally complemented with different viewing directions of the same crystal, or “thickness maps” from STEM-HAADF.

For XPS, I find the difference in binding energies for AS-TiO2 odd. I cannot find any details in ref. 18, which is cited as an explanation. On the contrary, ref. 18 shows identical binding energies for all anatase facet types investigated. The difference could be related to the impurities, or to a difference between anatase and rutile. If the latter, it would be good to include results from CM-TiO2 as a reference.

Furthermore, the authors claim no O-defects on line 191. Then, the strong (khaki) colouring of the samples needs an explanation. The presence of Ti3+ (i.e., O-defects) is one of the primary causes of this type of colouring in otherwise pure TiO2 (see e.g. Sci. Rep. 5, 10714; doi: 10.1038/srep10714 (2015)).

The difference between the two anatase samples (CM and AS) seems to be entirely explained by the specific surface area. But without a clearer statement of the purpose of the study, I cannot judge the significance of this.

Minor points:

The abstract contains a series undefined abbreviations. I cannot find what No- (or NO-) stands for at any point. This needs to be clarified.

A series of applications are mentioned in the intro, then the authors state on line 31: “…effectively realizing the above applications, as its photocatalytic activities depend critically on the crystal phase, morphology, size, surface area …”. But not all the listed applications are photocatalysis related.

A series of synthesis methods are described in detail in the introduction. What lies behind this selection, out of the large number of papers describing shape-selective TiO2 synthesis (see review mentioned earlier)?

The paper contains some claims that are not explained/substantiated: l.263 the activity of CM-TiO2 depends on the heterojunction structure between anatase/rutile, l.274 NO-TiO2 has the highest photodegradation amount due to having the lowest surface area.

L.355: the most muscular separation efficiency. What does muscular mean in this context?

Author Response

Please see the attachment"Response to Reviewers-catalysts-1550397.docx".

Reviewer 2 Report

See attached file

Author Response

Please see the attachment "Response to Reviewers-catalysts-1550397.docx".

Reviewer 3 Report

This is an interesting article that can be recommended for publication, but after clarifying and detailing some parts of the text.

  1. Line 16.  Abbreviations here and below need to be transcribed.
  2. Line 33 needs references, especially for TiO2(B).
  3. Line 38-45. More information about TiO2 and recent developments are needed here. See recent MDPI papers:

Serga, V.; Burve, R.; Krumina, A.; Romanova, M.; Kotomin, E.A.; Popov, A.I. Extraction–Pyrolytic Method for TiO2 Polymorphs Production. Crystals 202111, 431. https://doi.org/10.3390/cryst11040431

Dima, R.S.; Phuthu, L.; Maluta, N.E.; Kirui, J.K.; Maphanga, R.R. Electronic, Structural, and Optical Properties of Mono-Doped and Co-Doped (210) TiO2 Brookite Surfaces for Application in Dye-Sensitized Solar Cells—A First Principles Study. Materials 202114, 3918. https://doi.org/10.3390/ma14143918

and references therein.

  1. Line 42. Here, a clear description and explanation is needed, why, among all nanostructures, it is nanorodes that will be investigated.
  2. Line 68. “Experimental” paragraph is missing.
  3. 2 and 3. How stable is this data in the pictures over time
  4. 3. Were there any effects of aging/radiation induced transformation during TEM/HRTEM measurements?
  5. 5. If the photon energy is 325 nm (3.8 eV), then in this case electron-hole pairs are created, therefore a separate interpretation must be given for each luminescence band.
  6. Why were the excitation spectra not measured for each luminescence subband?
  7. Figure 7.  Irradiation time here is an incorrect value, because it depends on the intensity of the incident light.

Author Response

(The authors gave the same response as above.)

Round 2

Reviewer 1 Report

  1. Significance and novelty

Authors: Thank you very much for your comments. The synthesis method is novel. According to your suggestions, we added some sentences in the Introduction Section. “Although the preparation of microflowers, microspheres, and fan-shaped particles with different crystal forms and various exposed crystal facets using TiCl3 as raw material has been reported in the previous literature [21], the previous reports did not involve morphology-controlling agents such as AC, AS and urea. In this study, TiO2 crystals with different crystal forms, different morphologies and different exposed crystal planes were prepared by simply changing the type of morphology-controlling agent, which is innovative to a certain extent.”

Reviewer: Urea has been reported as a morphology controlling agent, see G. Liu et al. (Chem. Rev. 2014, 114, 9559), with very similar results as reported here. For the two ammonium salts (carbonate and sulfate) I have not found any direct examples in the literature, although other ammonium-based reagents have been used. These examples have not used TiCl3 as a raw material, but other Ti-compounds.

Separately, when using TiCl3 as the starting material, plenty of other morphology controlling reagents have been reported. Several are mentioned in the review (a quick search also yields 10.12720/ijmse.1.1.5-7). See below for more.

  1. Significance and novelty

Authors: Thank you very much for your comments. The novelty of this study is that with TiCl3 as titanium source, TiO2 particles with different crystal phases, morphologies and exposed crystal planes can be prepared by simply changing the type of morphology control agent.

Reviewer: I understand this to state that it is the specific combination of the TiCl3 precursor and these morphology controlling agents that forms the niche for the present study. If this can be explained more explicitly, motivated and contrasted with other reported synthesis strategies, I will increase the “novelty”and "interest"  aspect in my review.

  1. TEM analysis

Author: Thank you very much for your comments. The presence of (101) and (002) fringes indicates a [010] viewing direction, but says nothing about the facets present on the top/bottom of the crystal; these top/bottom facets need not be perpendicular to the viewing direction. We agree with this statement. However, according to the existence of two atomic planes (101) and (002) in Figure 3(f), the lattice spacing of 0.35±0.01 and 0.48±0.01 nm, respectively, and the interfacial angle is 68.3°, it can be inferred that the basal plane of nanorods mainly corresponding to the {010} crystal facets (see Refs. 23 and 24.). In views of this, we have revised the corresponding sentences in the manuscript. Thank you for your advice on extrapolating crystal surfaces from TEM.

Reviewer: The authors have not answered the original comment. To me, it looks like they have only restated the claim that (101) and (002) fringes in the images require {010} faceting on the top/bottom of the crystal. Note that I fully accept the authors’ assignment of the lattice planes based on “the lattice spacing of 0.35±0.01 and 0.48±0.01 nm, respectively, and the interfacial angle is 68.3°”. This does indeed show that there are (101) and (002) planes, and that the crystal is viewed in [010].

However, the {010} faceting simply does not follow from this observation: an anatase crystal bound by only {001} and {101} facets would look the same when viewed in [010]. See for example figure 10 in the review by Liu: here the TiO2 island clearly does not have {010} facets, but the HRTEM image still shows (002) and (101) fringes. The authors cite G. Zhang (Appl. Surf. Sci. 2017, 391, 228–235) and Y. Du (Molecules 2021, 26, 6031), but both these papers make the same mistake.

The claim of {010} facets must be supported or withdrawn, in my opinion, for the study to be valid.

5-6. XPS results

I think the authors have provided a sufficient and plausible explanation for their XPS results. However, a comparison with CM-TiO2 in terms of XPS and colour would still be very valuable to support the connection between peak shifts and Ti3+ content. I think the paper would be strengthened by including this data. Would it not be possible to wait a little while longer until after the winter break before publishing this report?

  1. Difference between AS and CM samples

Authors: Thank you very much for your comments. The crystal structures of AS-TiO2 and CM-TiO2 are different. AS-TiO2 is anatase, while CM-TiO2 is the mixed phase of anatase (96.8%) and rutile (3.2%). Therefore, the differences in photocatalytic activities between AS-TiO2 and CM-TiO2 cannot be entirely explained by the specific surface area.

Reviewer: But the authors themselves clearly show in figure 7c that the surface area is sufficient to explain the difference between CM and AS samples. The CM sample has only 3.2 % rutile, and there is no evidence included in the manuscript to show that the rutile and anatase are connected in a heterostructure in this sample. Thus, two essentially anatase samples are compared and found to perform equally on a per-area basis. It is possible that there are different effects leading to the same efficiency (maybe presence of rutile for CM and faceting for AS), but this cannot be concluded based on the presented results.

The high efficiency of the NO-TiO2 sample is also still mysterious. The authors nicely explain how this value appears due to division with a very small specific area. However, this does not explain *why* the amount of MB degraded isn’t similarly reduced. Rutile on its own is rarely very photocatalytically active, as I recall from previous reports. Especially when comparing NO and AC samples – these are the two outliers in MB degradation performance, despite both mainly consisting of rutile. Similarly, the difference to the Urea sample is also intriguing. If these points could be elucidated it would strengthen the interest and overall merit of the manuscript.

Minor points 8-12: nicely answered by authors.

Author Response

Dear Reviewer,

Please see the attachment below for our response to your comments.

Kind regards,

  Yi-en DU

Reviewer 2 Report

see attachment

Author Response

(The authors gave the same response as above.)

Reviewer 3 Report

The authors have significantly improved the manuscript and now it can be recommended for publication.

Author Response

Dear Reviewer,

Thank you very much for your comments on our manuscript!

Kind regards,

Yi-en DU